# Oral Health-Related Quality of Life in an Institutionalized Population with HIV+/AIDS in the Northern Region of Mexico

**DOI:** 10.3390/healthcare12131352

**Published:** 2024-07-06

**Authors:** Luis Alberto Gaitán-Cepeda, Nydia Alejandra Castillo-Martínez, María del Carmen Villanueva-Vilchis, José Román Chávez-Méndez, Ángel Gastón Peralta-Alegría, Jaime Paúl Ferré-Soto, Diana Ivette Rivera-Reza

**Affiliations:** 1Department of Oral and Maxillofacial Medicine and Pathology, Research and Graduate Division, Dental School, National Autonomous University of Mexico, Mexico City 04510, Mexico; dianrvr@fo.odonto.unam.mx; 2Microbiology Laboratory, Facultad de Ciencias de la Salud, Universidad Autónoma de Baja California, Tijuana 21100, Mexico; nydia.castillo@uabc.edu.mx (N.A.C.-M.); roman.chavez@uabc.edu.mx (J.R.C.-M.); gaston.peralta@uabc.edu.mx (Á.G.P.-A.); paul.ferre@uabc.edu.mx (J.P.F.-S.); 3Department of Oral Public Health, School of Superior Studies, National Autonomous University of Mexico, León 37684, Mexico; cvillanueva@enes.unam.mx

**Keywords:** oral health, related quality of life, HIV, AIDS, oral lesions

## Abstract

Approximately 39 million people worldwide live with human immunodeficiency virus (HIV), and antiretroviral therapy (ART) has improved life expectancy for these individuals, with quality of life (QoL) being a crucial aspect. However, there is limited information on oral health-related quality of life (OHRQoL) for institutionalized patients with HIV. This study used a cross-sectional design and included 43 residents of a non-governmental institution who had a confirmed HIV diagnosis and a history of intravenous drug use. The Spanish version of the Oral Health Index Profile-14 (OHIPsp) was used to assess the OHRQoL, with the 50th percentile serving as the cutoff for good or poor quality of life. All 43 patients had one or more oral lesions, with 44.1% having AIDS-related oral lesions (AROLs). Over half of the participants (48.8%) reported a poor OHRQoL, and females experienced worse quality of life in all dimensions compared to males. Subjects with AROLs were three times more likely to have poor OHRQoL than those without AROLs (*p* = 0.03; OR = 3.1 _IC 1.04–9.6_). These results highlight the need for a comprehensive treatment plan for patients with HIV that includes oral health, particularly for women living in precarious conditions or who are institutionalized. Improving oral health can significantly enhance quality of life.

## 1. Introduction

Human immunodeficiency virus (HIV) is one of the most significant pandemics that affects humans worldwide. Approximately 85.6 million people (64.8 million–113.0 million) have been infected with HIV since the start of the pandemic, and 40.4 million (32.9 million–51.3 million) people have died from AIDS-related illnesses since the start of the epidemic [1]. Despite efforts to control the pandemic, 39 million (33.1 million–45.7 million) people live with HIV; therefore, the eradication of the virus seems to be a distant goal [1]. The introduction of highly active antiretroviral therapy (HAART) in the mid-1990s led to a significant reduction in the morbidity and mortality associated with HIV infection [2] and an increase in life expectancy. HIV infection has been reclassified as a chronic disease. The objective of anti-HIV treatment now not only includes the control and prevention of HIV viremia and opportunistic infections, but also actions aimed at improving the quality of life (QoL) and overall well-being of HIV+ individuals. The increase in life expectancy of patients with HIV is accompanied by the need to improve their quality of life.

Health-related quality of life (HRQoL) is an individual’s subjective perception of their ability to perform essential activities and is influenced by their current health status [3]. This assessment reflects the impact of a disease and its treatment on an individual’s perception of well-being [4]. Oral health-related quality of life (OHRQoL) refers to the impact of oral conditions, lesions, or diseases on a subject’s functional, social, psychological, and financial aspects [5]. Therefore, oral health and QoL are essential for overall health and well-being [4]. Early clinical findings indicate that oral infections, notably oral candidiasis and hairy leukoplakia, are closely related to HIV infection [6]. Oral candidiasis and hairy leukoplakia are believed to have significant diagnostic and prognostic relevance [7,8,9], and oral candidiasis serves as an important clinical marker of immune system dysfunction in patients receiving antiretroviral therapy (ART) [9].

Some articles in the scientific literature (e.g., [10,11,12]) reported the impact of oral health on the quality of life of people living with HIV/AIDS (PLWHA). While HIV subjects have been reported to have poor OHRQoL, associated with the presence of oral lesions [10,11], other studies have found risk factors similar to the non-HIV population [12]. The data on the effects of oral health on QoL are scarce, and information remains inconclusive [13,14,15]. 

An important factor contributing to this apparent contradiction is the diversity of the PLWHA subpopulations [16]. Subpopulations of PLWHA exhibit diverse characteristics influenced by social, demographic, and economic situations, including treatment accessibility and adherence to antiretroviral therapy (ART). The northern region of Mexico, particularly Tijuana Valley, has a significant number of migrants, resulting in a significant floating population who may not have access to healthcare services. The objective of the present study was to investigate the impact of oral lesions on the OHRQoL of institutionalized adult patients with HIV undergoing ART in Northern Mexico. These data highlight their perceived need and will enable the establishment of preventive measures and interventions to improve the quality of life of subjects who are HIV positive. 

## 2. Materials and Methods

*Study Design and Subject.* This transversal study encompassed residents of a non-governmental institution (Las Memorias A. C, Tijuana, Baja California, Mexico), focusing on housing subjects, with a confirmed diagnosis of HIV infection and a history of intravenous drug use. All participants volunteered and received comprehensive explanations of the study objectives, scope, and procedures. Before the study, the participants signed an informed consent form that guaranteed the confidentiality and anonymity of the data and the option to withdraw from the study at any time. This study was conducted in November 2018. The research protocol was authorized by the Research and Bioethics Committee of Escuela de Ciencias de la Salud, Universidad Autónoma de Baja California (registration number 009/2018; September 2018).

*Inclusion criteria* required participants to have a confirmed diagnosis of HIV infection and a history of intravenous drug use to be able to complete the interview.

*Exclusion criteria* included those who did not provide complete information or had any impediments to completing the interview. Patients who were psychologically compromised, had neurological deficits, or refused to provide informed consent were excluded from the study. 

*Oral examination.* The oral examination was conducted by an expert in oral medicine (LAGC) who utilized all necessary protective measures, including a dental chair, dental mirrors, and artificial light. The examiner collected information on the presence of dental, AIDS-related oral lesions not related to HIV+/AIDS using clinical parameters. Caries was visually identified based on the World Health Organization criteria [17], where the presence of any cavitation associated with the presence of biofilm is registered as caries. On the other hand, as this was a field study in immunocompromised patients, the evaluation of the presence of gingivitis was purely visual, taking into account the general presence of inflammation, bleeding, and/or redness of the gingival tissues. 

*Oral health index profile and oral health-related quality of life.* After oral examination, all participants were interviewed in person using the Oral Health Impact Profile (OHIP-14sp). OHIP-14 is a widely validated instrument for assessing the impact of oral health on an individual’s quality of life [18,19,20]. The OHIP-14 version has been validated in Spanish. It is considered a comprehensive and reliable psychometric tool for assessing seven dimensions of OHRQoL: functional limitations, pain, psychological discomfort, physical disability, psychological disability, social disability, and incapacity [21]. Since its validation in 2009, the OHIP-14 has not been significantly modified and remains a valid and reliable measure of oral health-related quality of life. The OHIPsp uses a Likert-type scale ranging from 0 (never) to 4 (almost always) to evaluate the frequency of functional problems experienced in daily life owing to oral lesions. The responses were then coded and scored using the additive method, with a minimum possible score of 0 and a maximum score of 56. A higher score indicates a lower quality of life. The 50th percentile (median) was used as the cutoff point for good or bad quality of life in accordance with previous publications of our research group [22,23]. 

*Statistical analysis.* The statistical analysis included calculating the means and standard deviations for quantitative variables such as age and quality of life and percentages for qualitative variables. Student’s *t*-test (*p* < 0.05) was used to verify the difference in the mean OHIP-14 by sex. To confirm the association between good and poor quality of life with respect to sex and with respect to oral lesions, the chi-square test with a 95% confidence interval was used. Finally, the odds ratio value was obtained to determine the risk of having a poor quality of life according to the type of lesion present.

## 3. Results

*Clinical and demographical characteristics.* Forty-three institutionalized patients with HIV/AIDS (32 male [74%] and 11 women [25.6%]) were included in this study. The mean age was 44.7 years (standard deviation [SD] ± 9.55). The following oral conditions and lesions were diagnosed: regarding dental conditions, dental caries, residual dental roots, tooth loss, gingivitis, and periapical abscess were diagnosed; regarding *AIDS-related oral lesions* (AROLs), hairy leukoplakia, pseudomembranous candidiasis, erythematous candidiasis, and angular cheilitis were diagnosed; and regarding conditions *not related to HIV+/AIDS*, fissured tongue, saburral tongue, xerostomia, and melanotic macula were diagnosed. 

All 43 patients exhibited one or more oral lesions, and we did not observe any lesion-free subjects. A total of 43 (100%) subjects presented with dental alterations, 21 (49.8%) had residual roots, and 9 presented with gingivitis. As for dental lesions, tooth loss remained constant due to a prevalence of 100%, followed by caries with 86%; 47.6% of the subjects presented with residual dental roots, gingivitis was observed in 19% of subjects, and periapical abscess was observed in 9.5% of subjects. Regarding oral mucosal lesions not related to HIV/AIDS, two subjects presented with a saburral tongue, and two presented with a fissured tongue. AROLs were diagnosed as erythematous or pseudomembranous candidiasis in seven cases, hairy leukoplakia in four cases, and angular cheilitis in eight cases. Antiretroviral treatment-related conditions, namely xerostomia and melanocytic macules, were identified in this population with prevalence rates of 13.95% and 6.9%, respectively. The total number of oral lesions was 143. The number of lesions was greater than that in the subjects because all patients concomitantly presented with two or more lesions. 

*Oral health index profile results.* The mean OHIP-14 value was 28.4 (SD ± 12.35), ranging from 4 to 51. Females had a mean OHIP-14 value of 31.6 (SD ± 14.37), ranging from 11 to 51, while males had a mean value of 27.4 (SD ± 11.67), ranging from 4 to 47. The dimension with the highest OHIP-14 score was psychological discomfort (6.1) (SD ± 2.17), followed by functional limitation (5.23) (SD ± 2.39). Regarding gender, in the females, the dimensions with the greatest values and consequently the most significant impact on quality of life were psychological discomfort (6.2) (SD ± 2.24), followed by functional limitation (5.3) (SD ± 2.54) and pain (5.09; SD ± 2.21). In males, the dimensions with the highest scores were psychological discomfort (6.09; SD ± 2.19) and functional limitation (5.1; ± 2.38). The comparison of means found that women had a worse OHRQoL in the dimension of disability than males, with the difference being 4.4 vs. 2.7 (*p* = 0.048). Although the rest of the dimensions did not show statistical significance (Table 1), women reported worse QoL in all dimensions. 

*Oral health-related quality of life*. The participants were categorized into good and poor quality of life groups, with the 50th percentile (median) being the cutoff point. Thus, values >50 were classified as a poor quality of life, whereas values <50 were considered good. Of the total sample (n = 43), 21 patients (48.8%) were classified as having a poor QoL related to oral health. The chi-square test was used to identify whether there was an association based on sex; however, no dimension showed statistical significance, as shown in Table 2.

When relating poor QoL (n = 21) with AROLs, we found that six (28.6%) patients had angular cheilitis, five (23.8%) had candidiasis, and three (14.3%) had hairy leukoplakia. The association between AROLs and poor quality of life was statistically significant (*p* = 0.03; OR 3.1IC 1.04–9.6) (Table 3). Thus, subjects with AROLs were three times more likely to have a poor quality of life than subjects with HIV+/AIDS who did not have AIDS-related lesions.

## 4. Discussion

The purpose of this study was to determine the impact of oral health on the quality of life of institutionalized patients with HIV/AIDS under ART in the northern region of Mexico. Our results show that women had higher scores for all dimensions of the OHIP-14sp. The dimensions with the highest scores were psychological discomfort, functional limitation, and pain. This suggests a poorer QoL for institutionalized women with HIV/AIDS than for men with HIV/AIDS. A poor QoL is closely associated with AROLs. Subjects with HIV/AIDS who present poor oral health and are explicitly suffering from AROLs have a significantly worse OHRQoL than the HIV-seronegative population [10,13,14,15,24] and a worse QoL than that of women with HIV [25,26,27]. In our sample, the most affected dimension was psychological discomfort, which agrees with previous reports that place the worst values on the psychological dimensions [13,27,28]. The data mentioned above are relevant because psychological discomfort is a trigger or amplifier of anxiety and depression, which are common in subjects with HIV [13]. Our research group established an association between psychological discomfort and anxiety or depression in a previous study of patients with oral lichen planus (OLP) [23]. 

It is important to point out that all participants in the present study presented with dental problems. Dental caries was significantly associated with a poorer OHRQoL [26,29], and high decayed teeth index and dental loss are independently associated with OHIP scores [11]. Oral health programs are needed to improve the OHRQoL of PLWHA. Dental issues with the most significant impact are difficulty chewing food, pain when chewing, toothaches, and loose teeth. Some putative explanations for the high prevalence of dental problems in PLWAH have been proposed. On the one hand, this could result from a lack of interest or underestimation of its importance by patients. The study population had health priorities other than dental care. On the other hand, social and demographic factors such as employment and social and public health support have been associated with the OHRQoL of PLWHA [10,11,15,25,27,30]. The population in the present report had no jobs or governmental or social support, and the shelter did not have permanent dental services; therefore, they did not have access to dental treatment, including private dental treatment, because of a lack of income. Accordingly, active dental caries or sequelae of dental caries were present in 100% of the HIV subjects involved. 

To our knowledge, the present study is the first to examine the OHRQoL in institutionalized patients with HIV. There are few studies on the quality of life of (*no-institutionalized*) subjects with HIV in relation to oral health. Because different tools have been used to determine the impact of oral health on the quality of life of PLWHA, comparisons are complicated. Therefore, we reviewed the scientific literature and identified and isolated 11 papers that used OHIP 49/14 as a research tool (Table 4). Four used OHIP-49 [10,11,25,27], and seven studies used OHIP-14 [13,14,15,16,26,30,31]. However, even though OHIP-14/49 within a group of culturally homogenous individuals has high internal reliability, consistency, and validity [10], there are two methods to establish a poor or good quality of life: by using the percentage of responses to each question and by using the 50th quartile. Both methods are correct and offer interesting data but also prevent comparisons, deeper analysis, and meta-analysis. In addition, several articles only reported the total OHIP results without including the results by dimension. Homogenizing strategies to establish a cutoff point for good or bad quality of life would be desirable. As shown in the table, there is only one report on the institutionalized prison population. The remaining studies were conducted on outpatients. Therefore, the present study is the first to establish the impact of oral health on the quality of life of institutionalized PLWHA. Four articles that used the OHIP-49 as a research tool were identified [10,11,25,27]. The total value ranged from 14.4 in South African subjects without AROLs [10] to 123.3 reported in Portuguese subjects with HIV [11]. In the case of the results obtained using the OHIP-14 [13,14,15,16,26,30,31], the total value variation was also wide, ranging from 5.8 reported in Portugal [26] to 28.4 reported in the United States [15]. We obtained a total OHIP-14 value of 28.46, which places the institutionalized population of Northern Mexico at a higher score, meaning it has the lowest OHQoL compared to the published data. A study in Florida showed that subjects who were institutionalized or in prison had the highest score compared to subjects who were in their own homes, temporary housing, another person’s home, a drug treatment center, or even on the street [27]. 

This is the first study on the OHRQoL among institutionalized PLWHA on the northern region of Mexico. The majority of patients, before their inclusion in the institution, were homeless without economic resources or family support. Subjects in this situation are considered to have a high risk of HIV infection because the use of intravenous-recreational drugs is widespread in this population [32,33]. The “Las Memorias” shelter is a non-profit organization that houses individuals with a history of intravenous drug use and who are also HIV+; most of them have these experiences due to migration. The most frequent reason for all included subjects to be in the Tijuana Valley was migration to the USA. Tijuana, Mexico is the largest city on the Mexican–USA border in the state of Baja California, and it is situated on major migration. Migrants live in an important risk environment, including social networks; injection locations; population mobility; unavailable health services, including dental ones; inequities to ethnicity; and social stigma and discrimination [33]. Most subjects included in the present study were homeless at indeterminate times. Homelessness has been associated with a greater risk of HIV infection [34]. 

The present study has several limitations. We used a cross-sectional design with inherent methodological limitations as it is challenging to establish the correct temporal sequence of exposure and effect. We could not obtain the patients’ important social, demographical, and clinical data. Most patients did not know when or how they acquired the infection; we were unable to collect the infection times under HAART because although the included patients had been in the shelter for at least six months, they often did not remember whether they had previously undergone antiretroviral treatment, and if so, what kind of ART. Moreover, some patients needed an identity document. Regarding clinical aspects, there was a lack of objective measurements regarding dental caries and gingivitis/periodontitis; thus, we cannot assess the DMFT index or periodontal status. Although it is ideal to use an index, caries prevalence has also been used to indicate dental health status (Indu). The DMFT index is used to measure oral health. However, the DMFT index fails to take into account multidimensional measures of diseases (INDU) that are very complex in PLWHA. On the other hand, we would like to emphasize the value of having included the prevalence of both caries and gingivitis, which provides information on the oral health of the population, establishing that this prevalence was high and that the effect of this variable is homogeneous in the population studied, establishing the effect of the lesions themselves as a difference. 

The lack of a control group was a weakness of this study. Future studies on this population should take into account the particularity of the study group, which includes institutionalized patients who are HIV positive with a history of intravenous drug use and migrants, to find an adequate control group. We do not doubt that including a control group would have made the conclusions more solid and robust. 

Finally, the present study addresses the quality of life of PLWHA who are institutionalized, a particular subpopulation, so our results cannot be extrapolated to all PLWHA. 

## 5. Conclusions

Subjects with HIV/AIDS who are institutionalized in the northern region of Mexico, specifically women, have a poor OHRQoL. Our results highlight the need to establish a comprehensive treatment for PLWA that includes oral health, especially in subjects living with precarious conditions or who are institutionalized; improvements in oral health consequently improve quality of life. A poor quality of life is related to dental lesions and AROLs. Due to poor oral health and high treatment needs in the study setting, this population represents a vital priority group for the prompt delivery of dental care. More attention should be paid to common oral diseases, such as caries and periodontal diseases, where it is vital to establish routine dental check-ups for PLWHA. To further validate our findings, we propose a follow-up study (or cohort) to determine whether AROL treatment and dental rehabilitation improve QoL. 

Because human migration is not expected to decrease in the short or medium term, it is essential to know this population’s health conditions and dental health, which can profoundly affect their already precarious quality of life.

## Figures and Tables

**Table 1 healthcare-12-01352-t001:** Values of dimensions of OHIP-14 regarding gender.

DIMENSIONS	TOTAL (n = 43)	FEMALE (n = 11)	MALE (n = 32)	*p* *
Functional limitations	5.23 (±2.39)	5.36 (±2.54)	5.19 (±2.38)	0.83
Pain	4.58 (±2.18)	5.09 (±2.21)	4.41 (±2.18)	0.37
Psychological discomfort	6.14 (±2.18)	6.27 (±2.24)	6.09 (± 2.19)	0.81
Physical disability	3.28 (±2.49)	3.73 (±2.93)	3.12 (± 2.35)	0.49
Psychological disability	4.16 (±2.62)	4.27 (±3.32)	4.12 (± 2.39)	0.87
Social disability	1.88 (±2.36)	2.18 (±2.86)	1.78 (± 2.21)	0.63
Incapacity	3.19 (±2.48)	4.45 (±2.62)	2.75 (±3.31)	0.048
Total OHIP-14	28.46 (±12.36)	31.36 (±14.37)	27.47 (±11.67)	0.3735

± = standard deviation; *p* = statistical significance at 0.05; * = Student’s *t*-test.

**Table 2 healthcare-12-01352-t002:** Good and poor quality of life related to oral health regarding gender.

QUALITY of LIFE	FEMALE	MALE	TOTAL	OR (CI)	*p* *
	N (%)	N (%)	N (%)		
**FUNCTIONAL LIMITATIONS**					
GOOD	7 (65.6)	20 (62.5)	27 (62.8)	1.05 (0.21–5.95)	0.94
POOR	4 (36.4)	12 (37.5)	16 (37.2)		
TOTAL	11 (100)	32 (100)	43 (100)		
**PAIN**					
GOOD	5 (45.5)	17 (53.1)	22 (51.2)	0.73 (0.15–3.60)	0.66
POOR	6 (54.5)	15 (46.9)	21 (48.8)		
TOTAL	11 (100)	32 (100)	43 100		
**PSYCHOLOGICAL DISCOMFORT**					
GOOD	5 (45.5)	17 (53.1)	22 (51.2)	0.73 (0.15–3.60)	0.66
POOR	6 (54.5)	15 (46.9)	21 (48.8)		
TOTAL	11 (100)	32 (100)	43 (100)		
**PHYSICAL DISABILITY**					
GOOD	7 (65.6)	22 (68.7)	29 (67.4)	0.79 (0.16–4.61)	0.75
POOR	4 (36.4)	10 (31.3)	14 (32.6)		
TOTAL	11 (100)	32 (100)	43 (100)		
**PSYCHOLOGICAL DISABILITY**					
GOOD	6 (54.5)	18 (56.3)	24 (55.8)	1.16 (0.22–6.75)	0.83
POOR	5 (45.5)	14 (43.7)	19 (44.2)		
TOTAL	11 (100)	32 (100)	43 (100)		
**SOCIAL DISABILITY**					
GOOD	6 (54.5)	16 (50)	22 (51.2)	1.2 (0.25–6.1)	0.79
POOR	5 (45.5)	16 (50)	21 (48.8)		
TOTAL	11 (100)	32 (100)	43 (100)		
**HANDICAP**					
GOOD	4 (36.4)	18 (56.3)	22 (51.2)	0.44 (0.08–2.21)	0.25
POOR	7 (65.6)	14 (43.7)	21 (48.8)		
TOTAL	11 (100)	32 (100)	43 (100)		
**TOTAL**					
GOOD	5 (45.5)	17 (53.1)	22 (51.2)	0.73 (0.15–3.60)	0.66
POOR	6 (54.5)	15 (46.9)	21 (48.8)		
TOTAL	11 (100)	32 (100)	43 (100)		

% = percentage; OR = odds ratio; CI: confidence interval 95%; * = chi2 test; *p* = statistical significance at 0.05.

**Table 3 healthcare-12-01352-t003:** The relation of the presence of oral lesions to AIDS with good or poor quality of life.

ORAL AND DENTAL LESIONS		TOTAL N = 143	OHRQoL		*p*
			GOOD N = 80	POOR N = 73	
**DENTALS** **(N = 68)**	**CARIES**	**36**	18 (50%)	18 (50%)	>0.05
	TOOTH LOSS	43	22	21	>0.05
	RESIDUAL DENTAL ROOTS	21	11 (52.3%)	10 (47.6%)	>0.05
	GINGIVITIS	9	5 (55.5%)	4 (44.4%)	
	PERIAPICAL ABSCESS	2	0	2 (100%)	
	TOTAL	68	34 (50%)	34 (50%)	>0.05
**AIDS-RELATED ORAL LESIONS** **(N = 19)**	HAIRY LEUKOPLAKIA	4	1 (25%)	3 (75%)	
	ANGULAR CHEILITIS	8	2 (25%)	6 (75%)	
	CANDIDIASIS	7	2 (28.5%)	5 (71.4%)	
	TOTAL	19	5 (26.3%)	14 (73.6%)	**0.03;** **OR 3.1** **(CI 1.04–9.61)**
**ORAL LESIONS NOT RELATED TO HIV/AIDS** **(N = 13)**	FISSURED TONGUE	2	2 (100%)	0	
	SABURRAL TONGUE	2	2 (100%)	0	
	MELANOTIC MACULA	3	2 (66.6%)	1 (33.3%)	
	XEROSTOMIA	6	3 (50%)	3 (50%)	
	TOTAL	13	9 (69.2%)	4 (30.7%)	>0.05

OHRQoL = oral health-related quality of life; *p* = chi-square; OR = odds ratio; CI = confidence interval; candidiasis = pseudomembranous and erythematous.

**Table 4 healthcare-12-01352-t004:** Oral health-related quality of life in different countries determined using OHIP 14/49 as a research tool.

**AUTHOR/** **YEAR**	**COUNT**	**POPULATION**		**Oral Health Index Profile—49 Dimensions**							
			**N**	**TOT**	**1**	**2**	**3**	**4**	**5**	**6**	**7**
Yengopal and Naidoo (2008) [10]	SO AFR	N/S WITH AROL	71	82.12	15.52	18.15	15.53	12.54	10.38	4.18	5.99
Yengopal and Naidoo (2008) [10]	SO AFR	N/S WITHOUT AROL	79	14.4	4.94	6.15	1.35	1.16	0.50	0.11	0.19
Sánchez et al. (2011) [25]	ARG	AMBULATORY	200	83	14	17	14	11	11	7	8
Tomar et al. (2011) [27]	USA	AMBULATORY	594	53.7	11.6	10	11.8	7.3	5.6	3.9	3.5
Vasconcelos Moreira et al. (2021) [11]	POR	AMBULATORY	664	123.3	24.2	23.2	14.2	21.4	15.2	11.1	1.4
			**Oral Health Index Profile—14 Dimensions**								
**AUTHOR/** **YEAR**	**COUNT**	**POPULATION**	**N**	**TOT**	**1**	**2**	**3**	**4**	**5**	**6**	**7**
Mulligan et al. (2008) [15]	USA	AMBULATORY	597	28.4							
Santo et al. (2010) [26]	POR	AMBULATORY	101	5.83							
Busato et al. (2013) [16]	BRA	AMBULATORY	195	6.3							
de Quadros Coelho et al. (2015) [14]	BRA	AMBULATORY	422	12.29							
Mohamed et al. (2017) [18]	MAL	AMBULATORY	121	8.8	1.7	1.3	2.1	1.5	0.9	0.5	0.8
Umeizudike et al. (2021) [30]	NIG	AMBULATORY	352	8.05							
Umniyati et al. (2022) [31]	INDO	INMATE JAIL	24	11							
Umniyati et al. (2022) [31]	INDO	INMATE JAIL	30	17.5							
PRESENT STUDY	MEX	INSTITUTIONAL	43	28.46	5.23	4.58	6.14	3.28	4.16	1.88	3.19

Count = country; SO AFR = South Africa; ARG = Argentina; USA = United States of America; POR = Portugal; BRA = Brazil; MAL = Malaysia; NIG = Nigeria; INDO = Indonesia; MEX = Mexico; TOT = total value of OHIP; 1 = functional limitation; 2 = physical pain; 3 = psychological discomfort; 4 = physical disability; 5 = psychological disability; 6 = social disability; 7 = handicap.

## Data Availability

Data are contained within the article.

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
