# Peer review of "Oral Health-Related Quality of Life in an Institutionalized Population with HIV+/AIDS in the Northern Region of Mexico"

_healthcare, 2024, doi:10.3390/healthcare12131352_

Round 1
Reviewer 1 Report
Comments and Suggestions for Authors
Dear Authors,
In reviewing the manuscript numbered healthcare-3033511, entitled "Oral Health-Related Quality of Life in an Institutionalized HIV+/AIDS Population in the Northern Border of Mexico,"
The authors aimed to investigate the impact of oral lesions on the oral health-related quality of life (OHRQoL) of institutionalized adult HIV+ patients undergoing antiretroviral therapy (ART) in northern Mexico. However, there are some concerns about the study I may list them as follows:
1. In this study, it was shown that HIV-positive individuals exhibit poor quality of life related to oral health, particularly due to the presence of oral lesions. This result of the study is reiteration of a well-known fact. However, it is not clear that why the results of this study may contribute to our current knowledge and might be useful for further investigations.
2. The biggest limitation of the study is not to use DMFT index and screening of periodontal status. The reason why the authors did not take into effect the periodontal and oral health status of the patients is not clear.
The manuscript ought to revisit and revise once again and answer the above issues. However, it can be considered for publication.
Sincerely,
Author Response
We thank the reviewers for taking the time to review the manuscript and for their interesting and accurate suggestions and comments.
Q - In this study, it was shown that HIV-positive individuals exhibit poor quality of life related to oral health, particularly due to the presence of oral lesions. This result of the study is reiteration of a well-known fact. However, it is not clear that why the results of this study may contribute to our current knowledge and might be useful for further investigations.
A- As mentioned by the reviewer, the fact that oral health, specifically the presence of AIDS-related oral lesions, affects the quality of life of HIV-positive subjects has been previously reported. However, sociodemographic, economic, and healthcare access characteristics deeply influence the quality of life, well-being, and even the clinical evolution of people living with AIDS/HIV (PLWA), making it very difficult to consider PLWA as a single population. Each subpopulation has particularities. This is true for migrants who do not have access to health services. The population of HIV-positive migrants has not been studied in depth, even though migratory movements have increased in recent decades and do not seem likely to stop in the short- or medium-term. The results of the present study confirm what has been published for other PLWA subpopulations and raise questions about oral AIDS in a population that has not been studied to date. Additionally, it would be interesting to follow this population to study the impact of the oral lesions once the oral and dental rehabilitation is completed.
These considerations are included in the 3rd and 4th paragraphs, pp. 3, Introduction section; 1st paragraph, pp. 11, Discussion section; and Conclusions section, pp. 10.
Q- 2. The biggest limitation of the study is not to use DMFT index and screening of periodontal status. The reason why the authors did not take into effect the periodontal and oral health status of the patients is not clear.
A- We thank the reviewer for his valuable comments. Although it would have been interesting to have these data, we recognize that there was an important omission in measuring the indexes, which would have possibly provided more precise information on the dental health status of the population evaluated. However, we would like to emphasize the value of having included the prevalence of both caries and gingivitis, which provides information on the oral health of the population, establishing that this prevalence was high and; therefore, the effect of this variable is homogeneous in the population studied, establishing as a difference the effect of the lesions themselves. This consideration was included in the Discussion section 1st paragraph, p 10.
Q-3. Describes clinical methods for gingivitis.
The parameters have been added to the Materials and Methods section, 4th paragraph, p 3.
Reviewer 2 Report
Comments and Suggestions for Authors
The authors present a cross-sectional study with a small population subset, although of interest regarding the nature of the specific sociodemographic characteristics of this population (institutionalized, with a history of drug addition, migrants) living with HIV. There is a significant flaw which is the lack of a proper age-matched control group, so the study design is not cross-sectional but instead this is a case series study. Also, there are several studies on the quality of life related to oral health and HIV infection and this study does not add novel findings to this knowledge area. Instead the data is mostly presented as descriptive data, without significant inferences from it, based also on the fact that there is no control group.
Minor issues are listed below:
- The introduction and discussion do not include an important study on the topic published in 2021 ( 10.1080/09540121.2020.1798866) Please consider adding this reference.
The study was conducted in November 2018, why was it only being submitted for publication now?
The reason for considering the 50% percentile as the cutoff for the quality of life should be further justified.
- Please remove the following sentence from the Materials and methods section : "The Materials and Methods should be described with sufficient details... be briefly described and appropriately cited." Also remove the following sentence from the beginning of the results section : "This section may... that can be drawn."
Author Response
We thank the reviewers for taking the time to review the manuscript and for their interesting and accurate suggestions and comments.
Q.1-The authors present a cross-sectional study with a small population subset, although of interest regarding the nature of the specific sociodemographic characteristics of this population (institutionalized, with a history of drug addition, migrants) living with HIV. There is a significant flaw which is the lack of a proper age-matched control group, so the study design is not cross-sectional but instead this is a case series study.
A. From our point of view, we consider the design as a cross-sectional study because we examined both the exposure variable (HIV oral lesions) and the effect variable (oral health-related quality of life) at one point in time, without taking into account a previous diagnosis of the participants and, on the contrary, inquiring about the relationship between the presence of lesions and the dependent variable. Thus, by simply observing the natural distribution of these variables, the design did not consider the inclusion of a control group.
The lack of a control group was a weakness of this study. The particularity of the study group, HIV-positive patients with a history of intravenous drug use, migrants, and institutionalized patients, made the search for a control group extremely difficult. However, we do not doubt that including a control group would have made the conclusions more solid and robust.
We have discussed this study's weaknesses in the 2nd paragraph, p 10 Discussion section.
Q.2- Also, there are several studies on the quality of life related to oral health and HIV infection and this study does not add novel findings to this knowledge area.
A- As mentioned by the reviewer, the fact that oral health, specifically the presence of AIDS-related oral lesions, affects the quality of life of HIV-positive subjects has been previously reported. However, sociodemographic, economic, and healthcare access characteristics deeply influence the quality of life, well-being, and even the clinical evolution of people living with AIDS/HIV (PLWA), making it very difficult to consider PLWA as a single population. Each subpopulation has particularities. This is true for migrants who do not have access to health services. The population of HIV-positive migrants has not been studied in depth, even though migratory movements have increased in recent decades and do not seem likely to stop in the short- or medium-term. The results of the present study confirm what has been published for other PLWA subpopulations and raise questions about oral AIDS in a population that has not been studied to date.
These considerations are included in the 3rd and 4th paragraphs, pp. 3, Introduction section; 1st paragraph, pp. 11 Discussion section; and Conclusions section, pp. 10.
Q.3- Instead the data is mostly presented as descriptive data, without significant inferences from it, based also on the fact that there is no control group.
A. Our descriptive cross-sectional study has limitations. However, we believe that the inferences we developed are both useful and valid. To further validate our findings, we propose a follow-up study (or cohort) to determine whether AROL treatment and dental rehabilitation improve QoL. This would corroborate our results and strengthen the credibility of our study.
We have discussed this study's weaknesses in the 2nd paragraph, pp 10 Discussion section; Conclusions section, pp. 10.
Minor issues are listed below:
- The introduction and discussion do not include an important study on the topic published in 2021 ( 10.1080/09540121.2020.1798866) Please consider adding this reference.
A. The interesting article the reviewer recommends is included with reference number 11 and referenced several times in the introduction and discussion sections.
- The study was conducted in November 2018, why was it only being submitted for publication now?
A.- The study was conducted in November 2018, and the data processing and analysis were conducted in 2019. However, participating institutions suspended their activities as of March 2020 due to the COVID-19 pandemic. Therefore, writing and finalizing the manuscript for publication was interrupted. However, in the last four years in our country, there has been significant and massive migratory movement. Migration mainly occurs from South America, Central America, the Caribbean, and the country itself. Although imperative, this migrant population has not yet been studied. Therefore, the authors consider this a propitious moment to publish this study and provide valuable data for the attention of this population.
- The reason for considering the 50% percentile as the cutoff for the quality of life should be further justified.
A. Since the OHIP-14 instrument does not have author-defined cutoff points, the decision to set the cutoff point at the 50th percentile is left to the authors, based on our group's previously published experience. We have added the references in the Materials and Methods section.
- Please remove the following sentence from the Materials and methods section : "The Materials and Methods should be described with sufficient details... be briefly described and appropriately cited." Also remove the following sentence from the beginning of the results section : "This section may... that can be drawn."
A- The sentences above were deleted; I apologize for the error.
Reviewer 3 Report
Comments and Suggestions for Authors
Dear Authors,
your manuscript needs major revisions.
Find it attached.
Yours sincerely,
Reviewer

Dear Authors,
your manuscript requires minor revisions in English language.
Yours sincerely,
Reviewer
Author Response
Q-line 18 – Write the abbreviation ART in brackets. Write only the abbreviation each time. Please keep this in mind throughout the manuscript.
A.- The abbreviation was included, and the entire manuscript was revised.
Q-line 19 – The abbreviation QoL in brackets.
A.- The abbreviation was included, and the entire manuscript was revised.
Q-line 22 – Is the abbreviation OHIPmx spelt correctly?
A -It was corrected to OHIPsp.
*Keywords – Write according to Medical Subject Headings 2024.
A.- The keywords were written according to Medical Subject Headings 2024.
*Introduction
Q-line 61 – The sentence is too long. Rephrase it.
A- The sentence was rephrased.
*Materials and Methods
Q-line 77 – Delete the first paragraph. I think this is an accidental error.
A- The sentences above were deleted; I apologize for the error.
Q-line 88 – Inclusion and exclusion criteria should be clearly stated. Each in his own line.
A- The inclusion and exclusion criteria were stated in own line. 2nd and 3rd paragraph, pp 3, Material and methods section.
Q-line 96 – The clinical oral findings are not well written. I suggest splitting it into: Dental, AIDS related oral lesions, not related to HIV+/AIDS.
A. The suggestion was attended.
Instead, I suggest you that you insert a sentence from the Results (line 122).
A. The suggestion was attended to.
Q. You also need to write how you determined dental caries and gingivitis (methodology).
A. The methodology to determine dental caries and gingivitis was included in the 4th paragraph, p 3, Material and Method section.
Q-line 97 – The last sentence should be at the beginning, i.e. part of the first paragraph.
A. The suggestion was attended to.
Q-line 99 – For which period is the OHIP-14 scale valid. Please write this as well.
A. Since its validation in 2009, the OHIP-14 has not been significantly modified and remains a valid and reliable measure of oral health-related quality of life. This sentence was included in the 5th paragraph, pp3, Material and Method section.
Q-line 109 – Write a separate paragraph Statistical Analysis. And improve it.
A. The statistical analysis was describe in paragraph 6th, pp3, Material and method section. The follow sentence was added “Means and standard deviations were calculated for quantitative variables such as age and quality of life, and percentages were calculated for qualitative variables. A Student's t-test was used to verify the difference in mean OHIP-14 by sex. To confirm the association between good and poor quality of life with respect to sex and with respect to oral lesions, a chi-square with 95% confidence was used. Finally, the odds ratio value was obtained to determine the risk of having a poor quality of life according to the type of lesion present”.
Q-line 113 – This sentence should be part of the first paragraph. Also write the date of approval.
A. The suggestion was attended
Q. Materials and Methods are written quite confusingly. It is necessary to divide them into separate subheadings, e.g. Study Design and Subjects, Inclusion and Exclusion Criteria, Oral and Dental Status, OHIP-14, etc.
A. The suggestion was attended
*Results
Q-line 117 – Remove the first paragraph. I believe this is an accidental error
A. The sentences above were deleted. I apologize for the error.
Q-line 122 – This sentence is part of Materials and Methods, as already written.
A. In this case, the results refer to the lesions diagnosed in the patients' oral examinations, so if it is not an inconvenience, we would like to keep them on this site.
Q-Table 3 – There is some missing data, i.e. there is no data for erythematous candidiasis.
A. The data on candidiasis include both erythematous and pseudomembranous forms. This clarification was at the bottom of the table.
Q. In all Tables in the Results, you must indicate which statistical analyses you have used and provide a list of abbreviations (in alphabetical order) at the bottom of the table.
A. The statistical test used was indicated at the bottom of each table.
Q. The Results (as well as Materials and Methods) are written in a rather confusing way. It is necessary to divide them into subheadings.
A. The suggestion was attended
*Discussion
Q-line 187 – Write the abbreviation OLP in brackets.
A. The suggestion was attended
Q-line 203 – The first two sentences are a bit contradictory. In the second, I suggest putting "not institutionalised" in brackets.
A. The suggestion was attended
Q. Write recommendations for future studies at the end.
A. Recommendations for future studies were included in the Conclusion section, p 10.
*References
Q-Write according to the journal's propositions.
A. The suggestion was attended